

# Association of circulating branched-chain amino acids with risk of pre-diabetes: a systematic review and meta-analysis

Yu Wang[1], Jiang-Hong Xiao[1], Xi-Mei Zhang[1], Wen-Xiao Wang[2], Qiao Zhang[1], Yu-Ping Tang[1] and Shi-Jun Yue[1,2]

[1] Key Laboratory of Shaanxi Administration of Traditional Chinese Medicine for TCM Compatibility, Shaanxi University of Chinese Medicine, Xi'an, Shaanxi Province, China
[2] International Joint Research Center on Resource Utilization and Quality Evaluation of Traditional Chinese Medicine of Hebei Province, Hebei University of Chinese Medicine, Shijiazhuang, China

Corresponding authors
Yu-Ping Tang,
yupingtang@sntcm.edu.cn
Shi-Jun Yue, shijun_yue@163.com

## ABSTRACT

**Objective**. Recent investigations have looked at the systemic concentrations of branched-chain amino acids (BCAA), which they consider prospective indicators for health conditions and the likelihood of chronic diseases. To elucidate the equivocal link between systemic BCAA concentrations and prediabetes, our study undertook a meta-analytical investigation.

**Materials and Methods**. Electronic databases were comprehensively searched in April 2024, and the study quality evaluation relied on the Newcastle-Ottawa Scale (NOS). The $I^2$ statistic was used for heterogeneity assessment, and data analysis relied on Review Manager 5.4 and Stata 12.0. Standard mean difference (SMD) was used as the effect size to account for varying units of measurement across the included studies. Sensitivity assessment was instituted to evaluate result tenacity, and subgroup examinations were concomitantly carried out, with funnel plots, Egger's regression analysis, and Begg's rank-correlation methodology deployed to discern publication bias. PROSPERO registration (CRD42024572760) validates this review's protocol compliance.

**Results**. Meta-analysis was conducted on 15 studies that involved 3,849 participants. Most individuals were over 40 years old. The prediabetes (PreDM) group exhibited significantly elevated levels of valine (Val) (SMD = 0.29; 95% confidence interval (CI) [0.14–0.45]; $P = 0.0002$), leucine (Leu) (SMD = 0.34; 95% CI [0.18–0.49]; $P < 0.0001$), and isoleucine (Ile) (SMD = 0.24; 95% CI [0.15–0.32]; $P < 0.00001$) compared to controls. Affirming the soundness of the results, sensitivity analysis indicated the lack of significant publication bias.

**Conclusions**. This meta-analysis supports the hypothesis that circulating BCAA levels increase in PreDM, suggesting that measuring BCAA levels could be investigated as a potential biomarker for the diagnosis of PreDM and a target for its treatment.

## INTRODUCTION

Around 1/3 of U.S. adults and a staggering 720 million individuals worldwide are influenced by prediabetes (PreDM), which is situated as a pivotal stage between normal glycemic

control and diabetes. By 2,045, this figure is projected to reach one billion (*Echouffo-Tcheugui et al., 2023*). The elevating prevalence of diabetes is accompanied by the elevating prevalence of PreDM, which progresses to diabetes in approximately 10% of cases annually in the U.S. (*Echouffo-Tcheugui et al., 2023*). An evident evolution towards Westernized living patterns emerging in China, India, and like-minded nations, the diabetes rate has surged to 10% (*Wheeler et al., 2020*). In type 2 diabetes mellitus (T2DM) patients, elevated systemic concentrations of branched-chain amino acids (BCAAs) have been documented. Notably, compared to healthy counterparts, higher levels of isoleucine (Ile), a particular BCAA, are linked to PreDM (*Long et al., 2020*). Tethered to an augmented inclination for the initiation of both PreDM and T2DM could be elevated BCAA levels, as growing evidence compellingly attests (*Guasch-Ferré et al., 2016*).

Branched-chain amino acids (BCAAs) comprise valine, leucine, and isoleucine. They are a vital subset of essential amino acids, sourced from animal proteins, fish, eggs, cereals, and vegetables. BCAAs constitute roughly one-third of muscle protein's amino acids and half of mammalian dietary essentials (*Górska-Warsewicz et al., 2018*; *Zhang et al., 2018*). Taking enough BCAA crucially impacts protein synthesis and energy balance, thereby enhancing muscle mass and physical fitness (*Neinast et al., 2019*).

Circulating BCAA levels are influenced by dietary intake and BCAA catabolism. Complete BCAA catabolism involves many enzymatic steps, with most taking place within the mitochondria. As metabolic dysfunctions like insulin resistance and glucotoxic stress can diminish the activity of branched-chain α-keto acid dehydrogenase (BCKD), which is the rate-governing enzyme in the two-step BCAA catabolic pathway, the concurrent accumulation of BCAAs and branched-chain α-keto acids (BCKAs) happens. Both metabolites have been shown to induce pancreatic β-cell apoptosis, oxidative stress, and insulin signaling disruption, ultimately resulting in systemic inflammation and vascular pathology, which may cause multi-organ dysfunction (*Wang et al., 2022*). A recent study found that plasma BCAA levels gradually increased, coinciding with both the onset and progression of diabetes mellitus (*Minelli et al., 2016*), proving that elevated circulating BCAA levels resulted in higher diabetes incidence (*Guasch-Ferré et al., 2016*; *Long et al., 2020*). Regarding the levels of BCAAs and the risk of the onset of hypertension and hyperlipidemia, akin connections have been revealed (*Flores-Guerrero et al., 2019*; *White et al., 2018*).

Intense scholarly discourse surrounds the discoveries of a proliferating array of studies, which have scrutinized the link between blood BCAA levels and prediabetes events. In a case-control study by *Safari-Alighiarloo et al. (2024)* the PreDM and T2DM groups presented significantly elevated concentrations of Leu, Val, and tyrosine *versus* the normal glycemic tolerance group. Heightened BCAA levels among the T2DM and glucose impairment (IGT) populations, which altered insulin responsiveness and lipid profiles, were detected in a comprehensive six-year follow-up research (*Andersson-Hall et al., 2018*). In contrast, in a retrospective study in Asia, the PreDM group had lower plasma BCAA levels *versus* healthy individuals (*Gaike et al., 2020*). Another retrospective study conducted at the First Affiliated Hospital of Jinzhou Medical University demonstrated that individuals with normal glucose tolerance had elevated serum Val levels compared to individuals with impaired fasting

glucose (IFG) (*Li et al., 2021*). Multifarious factors, including the idiosyncratic attributes of each investigation, the heterogeneous health conditions of participants, and the variable mitigation of confounding factors, may engender the disparities evident across various research results. The correlation between BCAA levels and PreDM having not been definitively quantified heretofore, this investigation applied a systematic review and meta-analysis to exhaustively scrutinize it.

## MATERIALS & METHODS

### General guidelines

The present study scrupulously adhered to the PRISMA extension reporting guidelines and underwent pre-registration on the PROSPERO database (registration number: CRD42024572760; https://www.crd.york.ac.uk/PROSPERO/#register).

### Search strategy

Databases were comprehensively searched, which covered PubMed, Embase, the Cochrane Library, Web of Science (WoS), ClinicalTrials.gov, WanFang, Weipu, China National Knowledge Infrastructure (CNKI), and China Biology Medicine (CBM) databases from the earliest available online index year to April 2024, with no language restrictions. The retrieval strategies are detailed in (Supplementary Material 3). Briefly, the search terms included all synonyms for BCAA exposure, PreDM outcomes, and human species. Additionally, relevant references from the review literature were manually retrieved.

### Inclusion and exclusion criteria

Inclusion criteria: (1) human-related studies; (2) exposure to the continuous variables Val, Leu, or Ile in circulating samples; (3) participants aged $\geq$ 18 without being diagnosed with type 1 diabetes mellitus (T1DM) or T2DM; (4) inclusion of a PreDM group diagnosed according to International Diagnostic Guidelines; and (5) collected in the fasting state, all biological samples indicated a certain consistency, while the included studies refrained from any dietary or pharmacological interventions.

The following studies were excluded from the analysis: (1) Abstracts, editorials, conference abstracts or notes, disease mismatch or exposure mismatch, and reviews. (2) Scholarly dissertations and studies that neither registered nor could leverage the risk factor under scrutiny, did not comply with the inclusion stipulations, were immaterial, or reproduced the patient sample were omitted.

### Literature selection and data extraction

The literature search and screening were independently carried out by Y.W. and X.M.Z., who first removed irrelevant studies and then excluded noncompliant ones through full-text assessment. After completing the screening process, two researchers extracted data from each article into a standardized spreadsheet. Via deliberation with a third investigator (S.J.Y.), all disparities were addressed to attain a concordance. The pivotal elements we retrieved from each article encompassed the principal author's appellation, publication annum, research schema, specimen count, population demographics, average

participant age, gender ratio, outcome indices, biospecimen types acquired, and analytical protocols executed. When multiple PreDM subgroups were identified, a specific formula was adopted to combine the mean ± SD of the outcome indicators. For incomplete data, the corresponding author was contacted to obtain the necessary information.

### Assessment of quality

Leveraging the Newcastle-Ottawa Scale (NOS), study quality was rigorously assessed across three realms: selection soundness (rated 0–4), group congruity (rated 0–2), and outcome sturdiness (rated 0–3). Studies with a score of ≥6 are confirmed to have a high quality.

### Statistical analysis

Exploited for the analysis were Review Manager 5.4 and Stata 12.0, the programs, with continuous variables manifesting as the outcome parameters in this research and the standard mean difference (SMD) applied as the effect quantum to harmonize the disparate units of measurement among the enlisted studies. The results were combined and reported as an SMD, each with a 95% confidence interval (CI) for the effect sizes. Heterogeneity was quantified by the $I^2$ test, where values above 60% were deemed indicative of substantial heterogeneity. For studies demonstrating low statistical heterogeneity ($P > 0.1$, $I^2 < 50\%$), a fixed-effects model was invoked; conversely, when heterogeneity was substantial or indeterminate, a random-effects model was deployed.

In cases where the included literature did not report the mean ± SD values, these were calculated using Eq. (1) (*Altman & Bland, 2005*) and when the data were represented by median and interquartile spacing, the formulas in the study of *Wan et al. (2014)* were adopted for converting the data into mean ± SD values. For studies with multiple subgroups, results were aggregated using the Eq. (2) (*Higgins et al., 2023*).

The combined results were presented as forest plots, showing statistical significance at the 0.05 level. To discern the impact of biological sample modalities and geographical locales on the outcomes, carried out were subgroup analyses; to detect latent publication prejudice, employed were funnel plots with Egger's and Begg's assays; to evaluate conclusion steadiness, conducted was a sensitivity scrutiny—all for a comprehensive appraisal of the research findings.

$$SD = SE \times \sqrt{N} \tag{1}$$

$$SD = \sqrt{\frac{(N_1-1)SD_1^2 + (N_2-1)SD_2^2 + \frac{N_1 N_2}{N_1+N_2}\left(M_1^2 + M_2^2 - 2M_1 M_2\right)}{N_1 + N_2 - 1}}. \tag{2}$$

*Note: SD: standard deviation; SE: standard error; M: mean value; N: sample size.*

## RESULTS

### Literature selection process

We retrieved 6,263 documents from the database, as shown in the PRISMA chart (Fig. 1), and an additional six documents were identified through other sources. After using Endnote

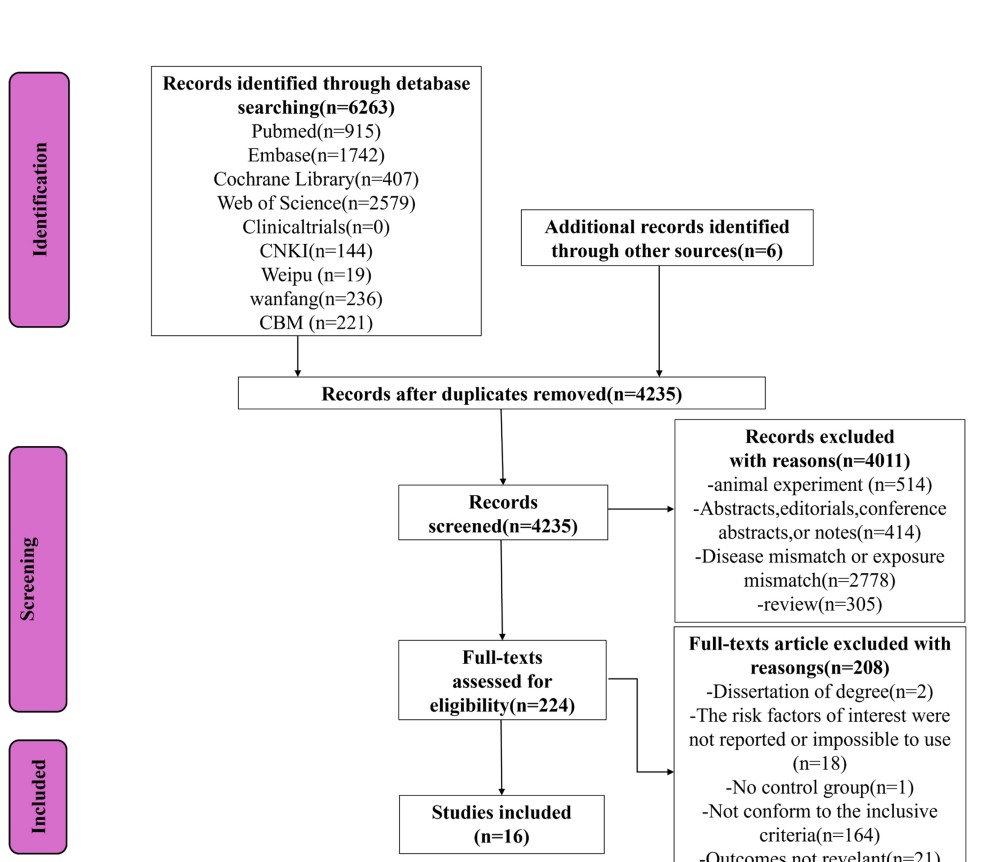

**Figure 1  PRISMA flowchart of the study.**

X9.1 software and manual review, 2,034 duplicates were removed. After reviewing the titles and abstracts, we excluded 4,011 of the remaining 4,235 documents. Following a full-text review, 208 documents were further excluded. Based on the inclusion criteria, 16 studies were included ultimately.

## Study characteristics and quality assessment

Two cross-sectional, 12 case-control, and two cohort studies were among the 16 studies, in which 1,478 PreDM individuals and 2,426 healthy controls were enrolled, and biological samples were serum in 10 studies and plasma in six. Seven investigations in Europe, seven in Asia, one in North America, and one in Africa; these scholarly endeavors were all executed across diverse regions. The sample sizes varied considerably, ranging from 31 to 855 participants. The risk factors of interest included the total concentrations of Val, Leu and Ile. The studies were published between 2010 and 2024, with specific characteristics

shown in Table 1. Based on the NOS quality scoring tool (Table S1), 10 publications were considered high quality, and six were deemed moderate quality.

## Meta-analysis
### Val

Data on Val levels were provided by a total of 15 studies, including 1,449 PreDM individuals and 2,394 controls (Fig. 2) (*Andersson-Hall et al., 2018*; *Cobb et al., 2014*; *Gaike et al., 2020*; *Hu et al., 2022*; *Kujala et al., 2016*; *Lee et al., 2018*; *Li et al., 2021*; *Mels et al., 2013*; *Menge et al., 2010*; *Mook-Kanamori et al., 2016*; *Owei et al., 2019*; *Safari-Alighiarloo et al., 2024*; *Shi et al., 2021*; *Tulipani et al., 2016*; *Xu et al., 2013*). Overall, the PreDM group had a significantly higher Val level *versus* controls (SMD = 0.29; 95% CI [0.14–0.45]; $P = 0.0002$), with substantial heterogeneity ($I^2 = 75\%$). Significantly disparate serum valine concentrations (SMD = 0.37; 95% CI [0.15–0.58]; $P = 0.0007$) were manifested by the PreDM group and controls based on the results of stratified analysis of biological samples, while the plasma valine concentration was statistically non-significant (SMD = 0.17; 95% CI [−0.07–0.40]; $P = 0.16$). Subgroup analysis by geographic region showed (Fig. 3) that Val levels were statistically significantly different between the PreDM group and controls in Europe (SMD = 0.22; 95% CI [0.06–0.37]; $P = 0.006$) and Africa (SMD = 0.33; 95% CI [0.13–0.53]; $P = 0.001$), whereas Val levels were not significantly different in Asian (SMD = 0.33; 95% CI [−0.05–0.72]; $P = 0.009$) and North American (SMD = 0.26; 95% CI [−0.07–0.59]; $P = 0.13$).

Figure S1 depicts the funnel plots meticulously devised by us to qualitatively appraise publication bias. Symmetrical funnel plots illustrated in Fig. S2 and the non-significant statistical outcomes of Egger's and Begg's tests ($P = 0.198$, $P = 0.235$) evidenced negligible publication bias. In further sensitivity analyses, excluding any individual study failed to greatly affect the results, reporting good stability (Fig. S3).

### Leu

Data on Leu levels were provided by 12 studies, including 1,015 PreDM individuals and 1,925 controls. (Fig. 4A) (*Andersson-Hall et al., 2018*; *Cobb et al., 2014*; *Gaike et al., 2020*; *Hu et al., 2022*; *Kujala et al., 2016*; *Lee et al., 2018*; *Li et al., 2021*; *Menge et al., 2010*; *Safari-Alighiarloo et al., 2024*; *Shi et al., 2021*; *Tulipani et al., 2016*; *Xu et al., 2013*). The PreDM group presented significantly higher Leu levels *versus* controls (SMD = 0.34; 95% CI [0.18–0.49]; $P < 0.0001$), with larger heterogeneity ($I^2 = 62\%$). Subgroup analysis of biological samples revealed that they presented remarkably different serum Leu levels (SMD = 0.38; 95% CI [0.19–0.57]; $P < 0.0001$) in a statistical aspect, while the difference in plasma Leu level was not statistically significant (SMD = 0.22; 95% CI [−0.13–0.57]; $P = 0.22$). According to the subgroup analysis of geographical regions (Fig. 5A), Leu levels were significantly higher in both European (SMD = 0.32; 95% CI [0.13–0.50]; $P = 0.0007$) and Asian (SMD = 0.33; 95% CI [0.07–0.60]; $P = 0.01$) PreDM group than in controls.

We drew the funnel plots for the qualitative assessment of the publication bias (Fig. S4). The funnel plot displayed signs of asymmetry, but Egger's and Begg's tests obtained non-significant results ($P = 0.290$, $P = 0.945$), suggesting no significant publication bias

**Table 1  Summary of the included studies.**

| Author-year | Study design | Population | Number | | Gender (M/F) | | Age | | Biological Sample | Detection method | Risk factors |
|---|---|---|---|---|---|---|---|---|---|---|---|
| | | | PreDM | Control | Case | Control | Case | Control | | | |
| Xu-2013 | Case-control study | Singaporean | 24 | 60 | 73%M | 50%M | 52.0 (10.7) | 38.1 (7.8) | Serum | Gas chromatography/ mass spectrometry and liquid chromatography/ mass spectrometry | ① ② ③ |
| Owei-2019 | Nested case-control study | African Americans and European Americans | 70 | 70 | Not reported | Not reported | 48.5 (8.23) | 47.6 (9.16) | Plasma | Injection tandem mass spectrometry | ① ④ |
| Li-2021 | Case-control study | Chinese | 32 | 54 | 78.13%M | 70.37%M | 41.69 (11.70) | 54.22 (13.33) | Serum | MS/MS | ① ② |
| Lee-2018 | Case-control study | South Korean | 69 | 73 | 43.5%M | 37.0%M | 57 (48–63) | 32 (24–50) | Serum | High performance liquid chromatography-tandem mass spectrometry | ① ② ③ |
| Hu-2022 | Cross-sectional study | Chinese | 334 | 380 | 36.2%M | 32.4%M | 59.38 (5.79) | 57.56 (5.79) | Serum | Hydrophilic interaction chromatography-tandem mass spectrometric method | ① ② ③ |
| Connelly-2017 | Case-control study | Netherlander | 25 | 30 | 44%M | 36.7%M | 56(9) | 52(9) | Plasma | Nuclear magnetic resonance | ⑤ |
| Andersson-Hall-2018 | Case-control study | Swede | 46 | 139 | 100%F | 100%F | 35 (6) | 34 (5) | Serum | NMR Spectroscopy | ① ② ③ |
| Shi-2021 | Case-control study | Chinese | 60 | 60 | 40%M | 45%M | 57.7 (4.2) | 56.4 (4.4) | Serum | UHPLC system coupled to a triple quadrupole mass spectrometer | ① ② ③ |
| Mook-2016 | Cohort study | Netherlander | 186 | 174 | 63.4%M | 48.3%M | 58.0 (46.0–65.0) | 55.5 (46.0–65.0) | Plasma | Biocrates AbsoluteIDQ™ 150 kit | ① ④ |
| Menge-2010 | Case-control study | German | 17 | 14 | 29.4%M | 50%M | 60.1 (8.9) | 57.0 (6.3) | Plasma | HPLC using an Eppendorf Biotronic LC 3000 Amino Acid Analyzer | ① ② ③ |
| Cobb-2015 | Cohort study | Europeans from 13 countries involved | 112 | 843 | 44%M | 46%M | 50 (8) | 47 (8) | Plasma | UHPLC-MS-MS | ① ② ③ |
| Kujala-2016 | Case-control study | Finns | 252 | 214 | Not reported | Not reported | 72.7 (5.9) | 71.6 (6.1) | Serum | NMR metabolomics platform | ① ② ③ |
| Gaike-2020 | Case-control study | Western region of India | 17 | 35 | 35.3%M | 48.6%M | 46 (9.6) | 37 (7.6) | Plasma | HPLC coupled with solvent delivery systems, autosampler, and photodiode array detector (all from Agilent 1100 series, Agilent Technology, Germany). | ① ② ③ |

**Table 1** (*continued*)

| Author-year | Study design | Population | Number | | Gender (M/F) | | Age | | Biological Sample | Detection method | Risk factors |
|---|---|---|---|---|---|---|---|---|---|---|---|
| | | | **PreDM** | **Control** | **Case** | **Control** | **Case** | **Control** | | | |
| Mels-2013 | Cross-sectional study | From the Kenneth Kaunda Education district in the North-West Province of South Africa | 178 | 225 | 61.2%M | 40.9%M | 46.7 (8.50) | 43.0 (10.1) | Serum | Electrospray ionisation tandem mass spectrometry | ① ④ |
| Tulipani-2016 | Case-control study | Spaniard | 33 | 31 | 39.4%M | 19.4%M | 47.0 (12.03) | 45.7 (13.58) | Serum | TSQVantage™ triple quadrupole mass spectrometer, LC-MS/MS, ESI-MS/MS | ① ② ③ ⑤ |
| Safari-Alighiarloo-2024 | Case-control study | Iranians | 23 | 24 | 65.2%F | 79.2F | 50.90 (10.30) | 42 (9.80) | Plasma | Liquid chromatography coupled with tandem mass spectrometry | ① ② ③ |

**Notes.**

Abbreviations: M, male; F, female; Val, valine; Leu, leucine; Ile, isoleucine; Xleucine, Xleu, xLeucine represents the sum of leucine and isoleucine; BCAAs, branched-chain amino acids.

①Val; ②Leu; ③Ile; ④Xleu; ⑤BCAAs; Data are means (SD), number (%) or medians (90% range), as appropriate for variable.

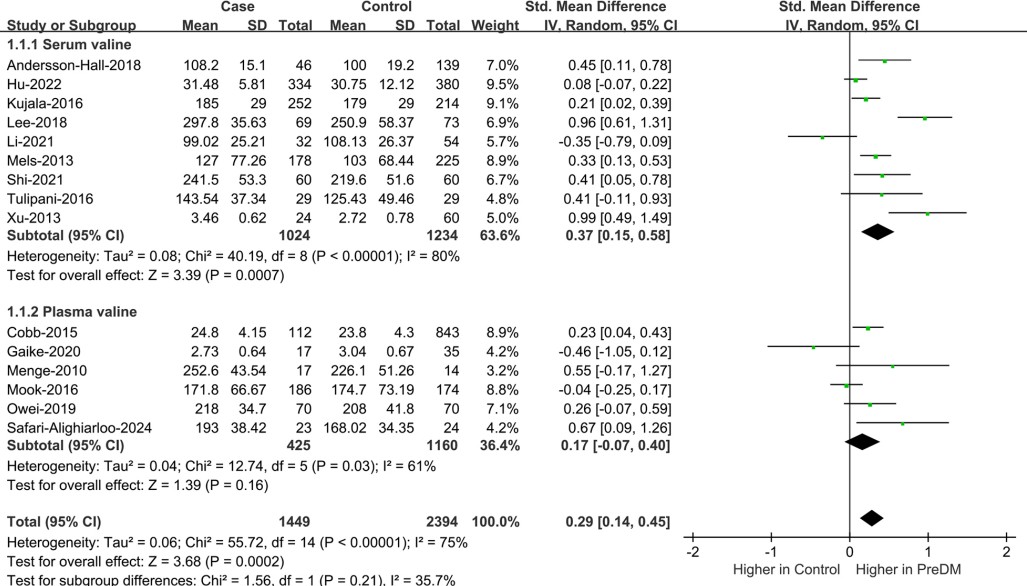

**Figure 2** **Forest plots of meta-analysis for assessing valine levels in PreDM and healthy.** PreDM, pre-diabetes; SMD, standard mean difference; CI, confidence interval; SD, standard deviation. Sources: *Andersson-Hall et al., 2018*; *Hu et al., 2022*; *Kujala et al., 2016*; *Lee et al., 2018*; *Li et al., 2021*; *Mels et al., 2013*; *Shi et al., 2021*; *Tulipani et al., 2016*; *Xu et al., 2013*; *Cobb et al., 2014*; *Gaike et al., 2020*; *Menge et al., 2010*; *Mook-Kanamori et al., 2016*; *Owei et al., 2019*; *Safari-Alighiarloo et al., 2024*.

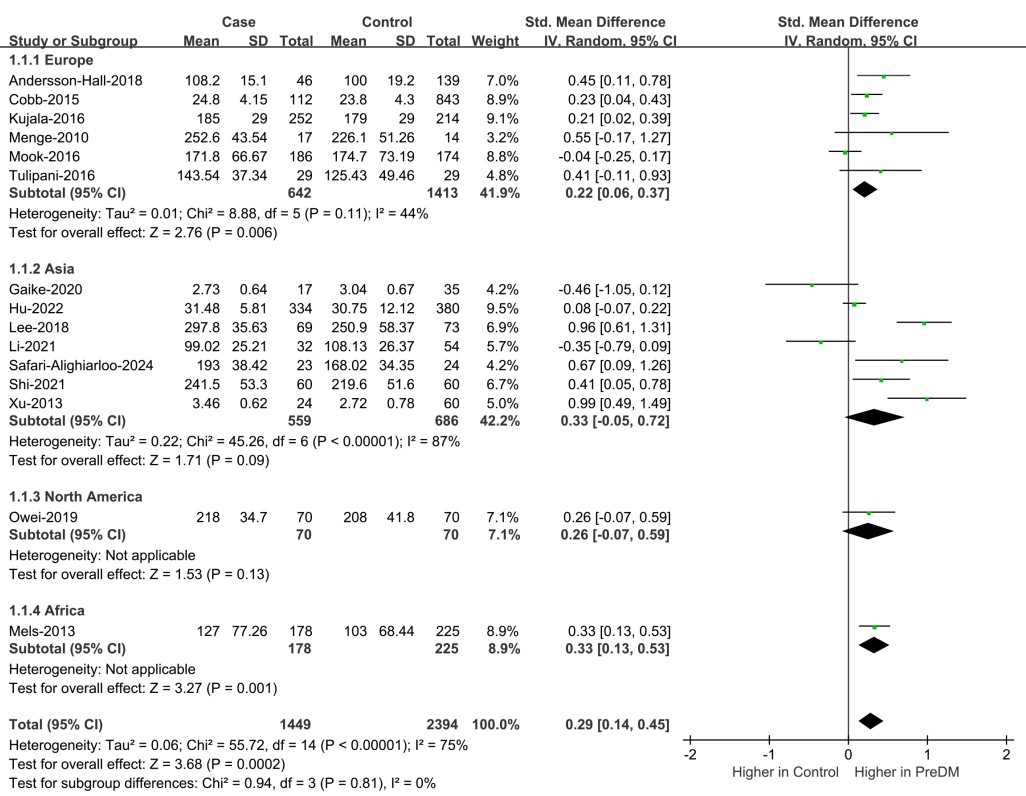

**Figure 3** **Forest plot of subgroup analysis to evaluate valine levels in PreDM and healthy groups in different geographical regions.** PreDM, pre-diabetes; SMD, standard mean difference; CI, confidence interval; SD, standard deviation. Sources: *Andersson-Hall et al., 2018*; *Cobb et al., 2014*; *Kujala et al., 2016*; *Menge et al., 2010*; *Mook-Kanamori et al., 2016*; *Tulipani et al., 2016*; *Hu et al., 2022*; *Lee et al., 2018*; *Li et al., 2021*; *Safari-Alighiarloo et al., 2024*; *Shi et al., 2021*; *Xu et al., 2013*; *Owei et al., 2019*; *Mels et al., 2013*; *Gaike et al., 2020*.

(Fig. S5). According to sensitivity analyses, excluding any individual study exerted no obvious impact on the results, proving the good stability (Fig. S6).

### *Ile*

Data on Ile levels were provided by 11 studies, including 983 PreDM individuals and 1,871 controls (Fig. 4B) (*Andersson-Hall et al., 2018*; *Cobb et al., 2014*; *Gaike et al., 2020*; *Hu et al., 2022*; *Kujala et al., 2016*; *Lee et al., 2018*; *Menge et al., 2010*; *Safari-Alighiarloo et al., 2024*; *Shi et al., 2021*; *Tulipani et al., 2016*; *Xu et al., 2013*). PreDM group exhibited significantly higher Ile levels *versus* controls (SMD = 0.24; 95% CI [0.15–0.32]; $P < 0.00001$), with low heterogeneity ($I^2 = 39\%$). Subgroup analysis of biological samples showed statistically different Ile levels in serum (SMD = 0.27; 95% CI [0.17–0.36]; $P < 0.00001$), but there was no statistical difference in plasma Ile levels (SMD = 0.13; 95% CI [−0.04–0.31]; $P = 0.13$). Subgroup analysis by geographical region showed (Fig. 5B) that Ile levels were statistically different between European (SMD = 0.22; 95% CI [0.10–0.34]; $P = 0.0003$) and Asian (SMD = 0.25; 95% CI [0.13–0.37]; $P < 0.0001$) PreDM group than in controls.

(A)

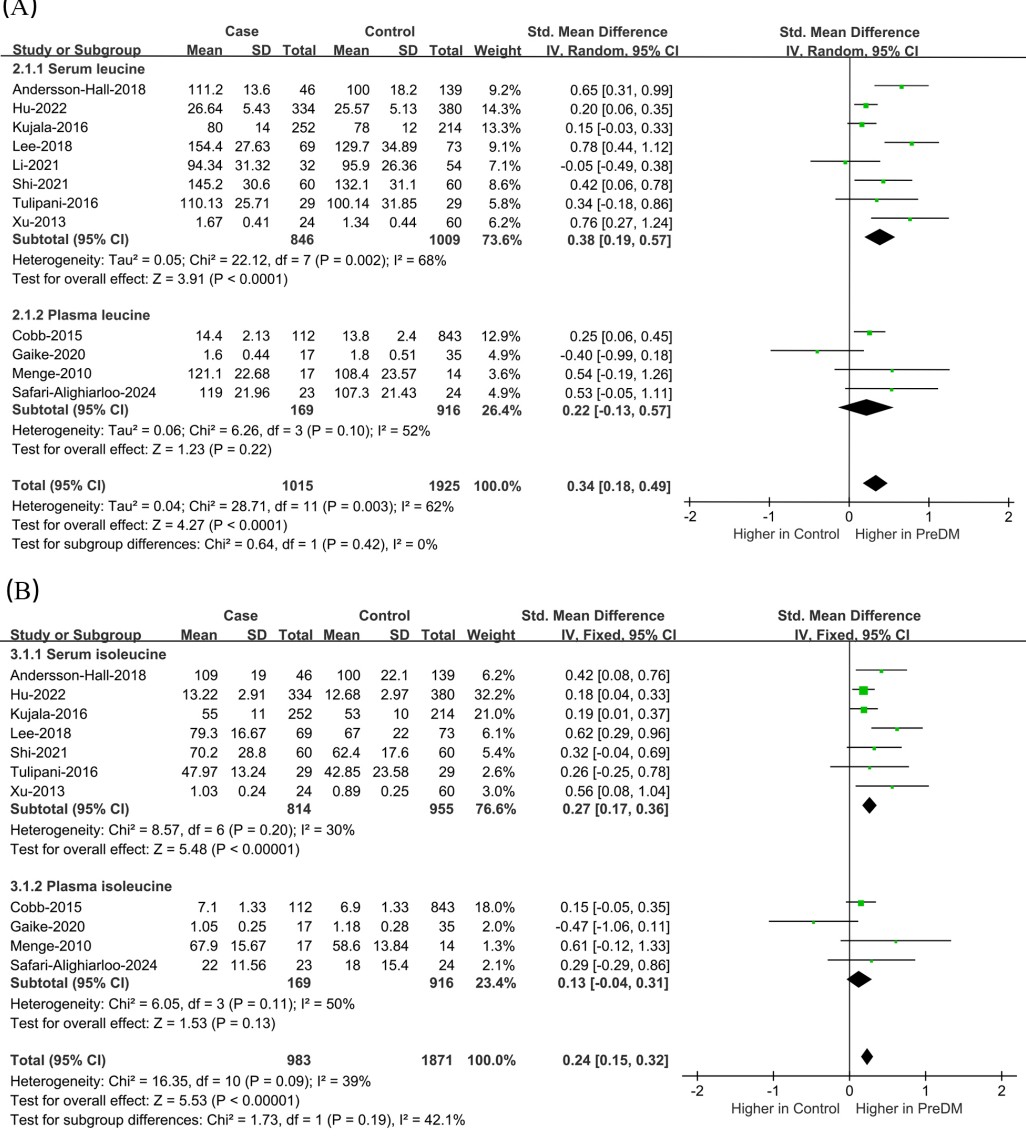

(B)

**Figure 4 Forest plots of meta-analysis for assessing (A) leucine levels and (B) isoleucine levels in PreDM and healthy.** PreDM, pre-diabetes; SMD, standard mean difference; CI, confidence interval; SD, standard deviation. Sources: *Andersson-Hall et al., 2018*; *Hu et al., 2022*; *Kujala et al., 2016*; *Lee et al., 2018*; *Li et al., 2021*; *Shi et al., 2021*; *Tulipani et al., 2016*; *Xu et al., 2013*; *Cobb et al., 2014*; *Gaike et al., 2020*; *Menge et al., 2010*; *Safari-Alighiarloo et al., 2024*.

Funnel plots were created for the qualitative assessment of potential publication bias (Fig. S7). The funnel plot displayed signs of asymmetry, with Egger's test and Begg's test obtaining non-significant results ($P = 0.341$, $P = 0.755$), demonstrating there was no presence of publication bias (Fig. S8). The validity of the study's inferences was substantiated by the sensitivity analyses depicted in Fig. S9, wherein the outcomes remained invariant irrespective of the individual study's exclusion.

(A)

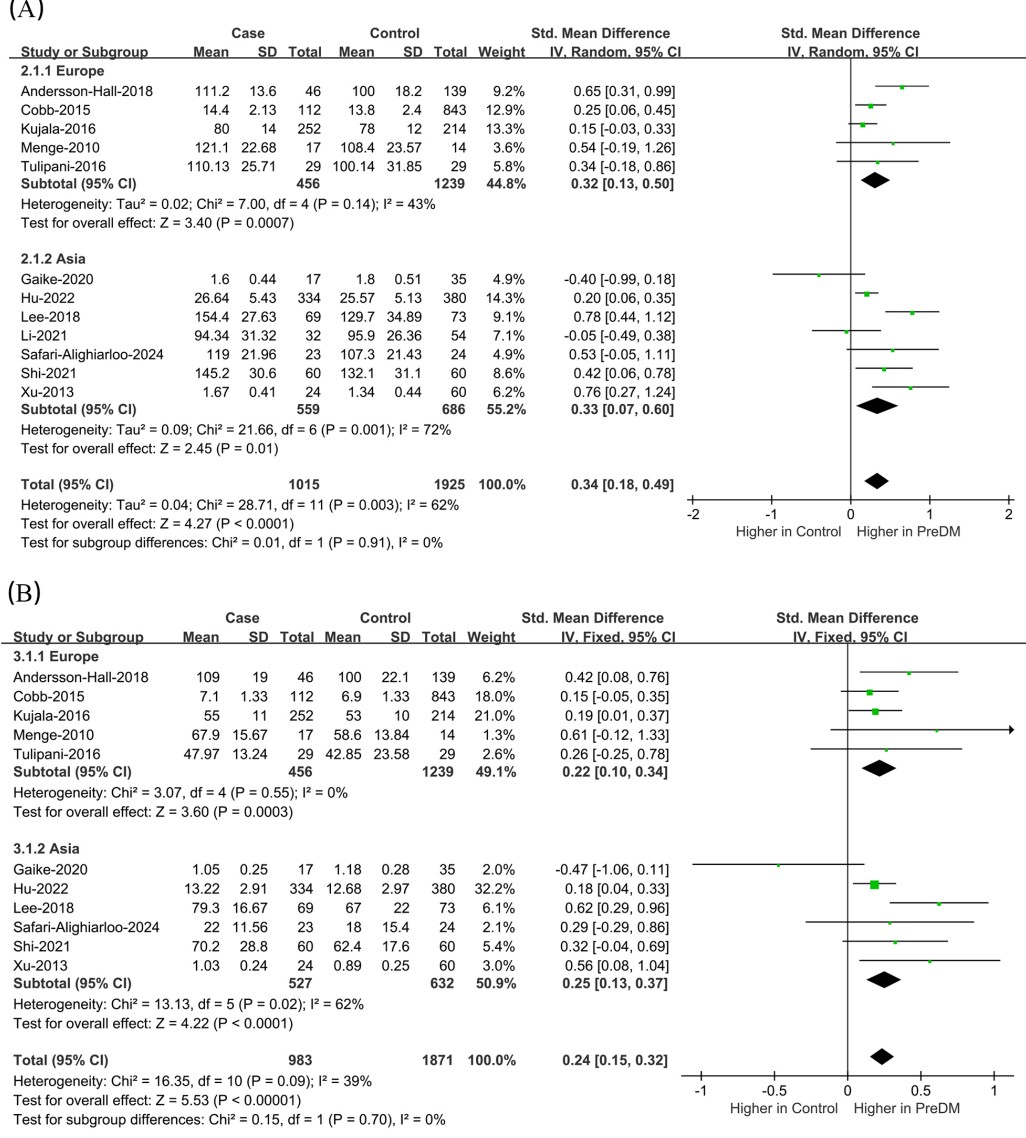

(B)

**Figure 5  Forest plot of subgroup analysis to evaluate (A) leucine levels and (B) isoleucine levels in PreDM and healthy groups in different geographical regions.** PreDM, pre-diabetes; SMD, standard mean difference; CI, confidence interval; SD, standard deviation. Sources: *Andersson-Hall et al., 2018*; *Cobb et al., 2014*; *Kujala et al., 2016*; *Menge et al., 2010*; *Tulipani et al., 2016*; *Gaike et al., 2020*; *Hu et al., 2022*; *Lee et al., 2018*; *Safari-Alighiarloo et al., 2024*; *Shi et al., 2021*; *Xu et al., 2013*; *Li et al., 2021*.

# DISCUSSION

Via undertaking a thorough systematic review and extensive meta-analysis incorporating a voluminous sample population, this scholarly work becomes the first-ever study to intently scrutinize the correlation between circulating BCAA amounts and PreDM. According to current evidence, the PreDM group possessed obviously higher circulating levels of Val, Leu, and Ile *versus* the healthy group. The combined analysis of circulating Val

and Leu showed considerable heterogeneity, necessitating careful interpretation of the findings. Subgroup analyses by biospecimen type and geographic region revealed some inconsistencies, highlighting the need for further studies to investigate the underlying causes. In conclusion, these findings imply that variations in circulating BCAA levels could be a useful reference for diagnosing and treating PreDM in clinical settings, potentially preventing the progression to T2DM, reducing the incidence of diabetic complications, and offering a therapeutic target for patients with PreDM and T2DM.

Recent studies have highlighted potential roles for BCAA clinically, particularly as biomarkers for the identification of risk factors linked to PreDM and therapeutic targets for disease prevention and treatment. For instance, a prior meta-analysis showed that higher circulating BCAA levels positively resulted in a higher risk of T2DM and PreDM (*Guasch-Ferré et al., 2016*; *Long et al., 2020*) and promoted the development of PreDM (*McCann et al., 2019*) and diabetes mellitus (DM) (*Rivas-Tumanyan et al., 2022*) and other risk factors. Additionally, adverse associations between elevated circulating BCAA levels and other PreDM risk factors have been extensively reported, including obesity (*Menni et al., 2016*), hypertension (*Flores-Guerrero et al., 2019*), dyslipidemia (*White et al., 2018*), and glycated hemoglobin A1c (HbA1c) (*Chung et al., 2021*). Circulating BCAA concentrations can be modulated *in vivo* by adipose tissue, which accomplishes this feat through the synchronized expression of pertinent metabolic enzymes within its cellular matrix (*Herman et al., 2010*). Salutary impacts on the metabolic well-being of young, developing mice, encompassing heightened glucose tolerance, a moderate diminution in adiposity accretion, and a rapid reversal of diet-induced obesity, were documented in another investigation for a diet expressly deficient in BCAAs (*Cummings et al., 2018*). BCAA regulates glucose and lipid metabolism primarily through the phosphatidylinositol 3-kinase/protein kinase B (PI3K-AKT) signaling pathway (*Liu et al., 2017*). Pathway analyses have shown that Ile promotes glucose uptake *via* PI3K independently of mammalian target of rapamycin (mTOR) signaling (*Doi et al., 2003*). By reducing the levels of glucose in the bloodstream and promoting the absorption of glucose by skeletal muscle, Ile moreover averts the progression of visceral obesity and hyperinsulinemia, thereby fostering overall metabolic well-being. (*Doi et al., 2005*; *Nishimura et al., 2010*). In oral glucose tolerance tests, Ile was more effective than Leu and Val in reducing plasma glucose levels in normal rats (*Doi et al., 2003*). Beneficial to glucose and lipid homeostasis is the reduction of BCAA and BCKA levels, which can be realized through two means: a diet restricted in BCAAs or the activation of BCKDH (*White et al., 2021*). The manipulation of the BCAA catabolic route, as evidenced by laboratory and animal experiments, affects crucial metabolic processes associated with glucose equilibrium. A highly effective strategy to re-establish glucose balance in metabolic conditions such as obesity and type 2 diabetes, as demonstrated by persuasive evidence from rodent concept-validation studies, supplemented by limited human data, could be enhancing BCAA breakdown, despite the ongoing uncertainties in understanding tissue-specific BCAA metabolism in humans (*Vanweert, Schrauwen & Phielix, 2022*).

BCAAs are substrates for nitrogen-containing compound synthesis and signaling molecules for the regulation of glucose, lipid, and protein anabolism. Additionally,

they modulate intestinal health and immune function by virtue of specific signaling networks, especially the PI3K/AKT/mTOR signaling pathway (*Nie et al., 2018*). Postulated by Lynch et al. are two mechanisms, the initial one of which contends that the activation of mTORC1 signaling stemming from elevated dietary BCAA levels could precipitate the advancement of insulin resistance and type 2 diabetes mellitus (T2DM). Gleaned from in-depth investigations into maple syrup urine disease (MSUD) and organic aciduria, the second proposed mechanism posits that the elevation of BCAA levels serves as a sentinel biomarker for metabolic dysregulation, particularly evident in animal models and human patients afflicted with disorders disrupting BCAA catabolic pathways. Evidently linked to insulin resistance and T2DM is the mitochondrial impairment in β-cells and other tissues, which stems from the accretion of noxious metabolites arising from BCAA metabolic dysfunctions (*Lynch & Adams, 2014*). Scientists suggest that accelerated BCAA catabolism could boost gluconeogenesis and cause glucose intolerance. By enabling the conversion of glutamate to alanine *via* transamination, this process disrupts the body's metabolic stability (*Newgard et al., 2009*). Insulin resistance and T2DM, as detrimental metabolic health outcomes, are firmly linked to the heightened levels of circulating BCAAs that are commonly present in obese individuals (*Lynch & Adams, 2014*; *Nakamura et al., 2014*). The increased propensity for the development of cardiovascular disease (CVD) is directly linked to elevated initial BCAA levels, as revealed by *Ruiz-Canela et al. (2016)*. Orchestrating metabolic mechanisms in diverse pathological states, BCAAs indicate that the targeted regulation of their metabolic flux may represent a promising interventional strategy for disorders like PreDM.

It remains to be investigated whether Val, Leu, or Ile exert independent effects on PreDM or if additional specific mechanisms are at play. The findings from our meta-analysis are limited to reflecting the BCAA status in PreDM; they do not clarify whether PreDM affects BCAA levels or if elevated BCAA levels contribute to PreDM. In subgroup analyses based on biological samples, the results were inconsistent. For example, individuals with PreDM exhibited significantly elevated serum levels of Val, Leu, and Ile; however, these associations were not observed for plasma Val and Leu. Additional research is necessary to investigate these associations across various biospecimen groups and establish the causality of these relationships, aiming to resolve the inconsistent findings. During the regional-based subgroup investigation, no notable divergence in valine levels among the PreDM and control populations in Asia and North America was ascertained, thus highlighting the inconsistent outcomes. Despite these inconsistencies, our findings support the view that BCAA levels are generally higher in the PreDM group compared to healthy individuals. Additionally, we excluded some studies involving diet, as higher metabolite intake may artificially increase BCAA levels in metabolomics (*Wang et al., 2011*).

Nevertheless, this study had multiple strengths, notably the use of a comprehensive and systematic approach that significantly reduced the chances of missing or excluding important published data. The study population included participants from various countries and regions, reducing racial differences and providing a more accurate reflection of the PreDM condition in humans. Our outcomes were devoid of publication bias, as validated by the funnel plot, Egger's test, and Begg's test. In addition, the sensitivity

analyses offered compelling evidence regarding the robustness of these findings. Before offering recommendations, the study's limitations shall be primarily considered. First, not all relevant studies could be retrieved from certain databases by specified search terms. Second, there may be a publication bias favoring positive results, potentially introducing bias into the analysis. Third, as with other observational studies, it was not possible to fully control for undetected and residual confounding factors. Fourth, the included studies exhibited a moderate to high level of heterogeneity. Some sources of heterogeneity, such as education level, data collection methods, and the handling and storage of blood samples, were difficult to discern. Additionally, inconsistencies in the diagnosis of PreDM and variations in quality control calibration of metabolomics analyses across studies may have contributed to the heterogeneity. Future research should be designed to account for and provide more detailed reporting on these factors.

## CONCLUSIONS

A total of 16 studies were included in this study, of which 15 studies were included in the meta-analysis. According to the 15 studies in the systematic review and meta-analysis, the PreDM group presented a higher circulating BCAA level, suggesting that circulating BCAA could be used as a supplementary diagnostic tool for PreDM. However, whether the alteration in BCAA levels is a consequence of the pathogenesis itself or the result of other causative factors has not been clearly explained. Such mechanisms shall be highlighted in further basic research. Required to elucidate the underlying causes of these disparities are additional investigations, as certain inconsistencies were discerned in the subgroup analyses categorized by biological specimen type and geographical region, as previously mentioned. This study aims to provide valuable insights for clinicians managing prediabetes and for researchers investigating metabolic biomarkers for early detection. By synthesizing evidence comprehensively, it offers guidance for academic research, practical healthcare strategies, and policymaking.

### Funding

This research was funded by the Key Research Project of Yan-Zhao Medical Studies (YZZZ2024016), Scientific Research Projects of Higher Education Institutions in Hebei Province (QN2025158), and the University-level Research Project of Shaanxi University of Chinese Medicine (2023GP24). The funders had no role in study design, data collection and analysis, decision to publish, or preparation of the manuscript.

### Grant Disclosures

The following grant information was disclosed by the authors:
Key Research Project of Yan-Zhao Medical Studies: YZZZ2024016.
Scientific Research Projects of Higher Education Institutions in Hebei Province: QN2025158.
University-level Research Project of Shaanxi University of Chinese Medicine: 2023GP24.

## Competing Interests

The authors declare there are no competing interests.

## Author Contributions

- Yu Wang conceived and designed the experiments, performed the experiments, analyzed the data, prepared figures and/or tables, authored or reviewed drafts of the article, and approved the final draft.
- Jiang-Hong Xiao analyzed the data, prepared figures and/or tables, and approved the final draft.
- Xi-Mei Zhang performed the experiments, analyzed the data, prepared figures and/or tables, and approved the final draft.
- Wen-Xiao Wang analyzed the data, prepared figures and/or tables, and approved the final draft.
- Qiao Zhang analyzed the data, prepared figures and/or tables, and approved the final draft.
- Yu-Ping Tang conceived and designed the experiments, authored or reviewed drafts of the article, and approved the final draft.
- Shi-Jun Yue conceived and designed the experiments, authored or reviewed drafts of the article, and approved the final draft.

## Data Availability

This is a systematic review/meta-analysis.

## Supplemental Information

Supplemental information for this article can be found online at http://dx.doi.org/10.7717/peerj.20054#supplemental-information.

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
