# Peer review of "Association of circulating branched-chain amino acids with risk of pre-diabetes: a systematic review and meta-analysis"

_PeerJ, doi:10.7717/peerj.20054_

## Round 0.1 · original submission · Major Revisions

Both reviewers generally agree that the manuscript is well-written and provides a comprehensive analysis. However, Reviewer 1 suggests several clarifications, including revising the exclusion criteria, correcting some statistical language, and improving the focus of the discussion, especially regarding BCAA’s role in glucose metabolism and prediabetes. Reviewer 2 requests further clarification on the inclusion of Type 2 diabetes patients and suggests rewording some sentences for clarity, particularly around BCAA’s effects. Both reviewers also emphasize the need for clearer explanations in certain sections of the manuscript.

Reviewer 1 ·

Basic reporting

The manuscript titled “association of circulating branched-chain amino acids with risk of pre-diabetes: a systematic review and meta-analysis” is a meta-analysis study to elaborate if there has a difference in serum/plasma BCAA (Val, Leu or Ile)levels between healthy people and preDM. In this study, the authors conducted a though comprehensive literature search of 9 databases and finally enclosed 16 studies. After information extraction and synthesis, they found that both plasma and serum BCAA (Val, Leu or Ile)levels in preDM were higher than those in healthy people. The authors also carried out sensitivity analysis and publication bias analysis. The study proved that preDM is associated with increased BCAA, which may imply potential research directions of research for clinical researchers, and has practical significance. The conclusion obtained is reliable with proper data analyzing. However, the language expression of part of the content is not clear, and some sentences are difficult to understand.

Experimental design

The following issues still need to be mentioned:
1. Regarding the exclusion criteria(SECTION 2.3), exclusion criteria is carried out in the studies that have been included, so the content of Line 117-119 is confusing, can you reorganize the exclusion criteria, such as: language of literature, type of literature...
2, When the 95% confidence interval of SMD covers 0, or the P-value is greater than 0.05, the correct statement is: the difference between A and B is not statistically significant, and it cannot be expressed as the difference between A and B is small/little; When the 95%CI includes 0, or P-value is less than 0.05, it is also inappropriate to state that the difference between A and B is significantly different. There are many improper expressions in the text, which need to be modified (as in Line183-192,and so on…)
3. For meta-analyses with less than 9 included studies, it is not necessary to draw funnel plots, nor to conduct Egger's and/or Begg's tests
4. Results 3.1, 3.2 and 3.3 have detailed contents of Val, Leu and Ile, including subgroup analysis (about sample types and regions) and sensitivity analysis, publication bias analysis also been carried out. However, for Xleu and BCAA, the number of included studies is less than 4, I’d like to suggest the authors to abandon these two parts to make the manuscript more compact and persuasive.
5. The Discussion part is suggested to focus on the relationship between BCAA, glucose metabolism and preDM, irrelevant content such as BCAA and thrombosis can be omitted.

Validity of the findings

no comment

Additional comments

1、 Line124-126 “In cases of discrepancies, a third researcher (S.J.Y.) was consulted to reach a consensus” what does it mean? The third one makes the decision or three persons discuss to get a consensus?
2、 It seems inappropriate to describe the individual of preDM as patient (for example, Line181, Line183, Table1…). It is suggested to use preDM and healthy, not patient and control.
3、 Line128-131“When multiple PreDM subgroups were identified, a specific formula was adopted to combine the means and standard deviations (mean ± SD) of the outcome indicators.” This sentence is hard to understand. Can you give a reasonable explain?
4、 Line204-207 “According to the subgroup analysis of geographical regions (Fig. 5A), Leu levels were significantly higher in both European (SMD = 0.32; 95% CI = 0.13–0.50; P = 0.0007) and Asian (SMD = 0.33; 95% CI = 0.07–0.60; P = 0.01) prediabetic patients and controls.” This sentence is hard to understand.
5、 Lin 266 “Circulating BCAA levels overall were not statistically significant.” This is also a confusing sentence.

·

Basic reporting

Overall, the manuscript is clear and well-written. Proposed objectives have been reached and conclusions are in accordance with the findings.
Three points that need clarification:
1- In the inclusion and exclusion criteria authors state that type 1 DM patients were excluded. Do these studies also include type 2 DM, besides the PreDM group? If so, the comparision was only between PreDM and controls? Were type 2 DM patients excluded from your analysis?
2- Starting in line 299: "BCAA is frequently reported to mediate anti-obesity, anti-diabetic, and anti-insulin resistance effects, and the extensive body of research supporting these findings cannot be listed comprehensively. According to these studies, BCAA metabolism change could be a potential therapeutic target specific to PreDM across different scenarios". I believe this sentence should be rewritten for clarity. From the context I suppose that you meant that BCAA has deleterious effets on the comorbidities, but the way it is written it seems that BCAA has beneficial effects in obesity, diabetes and insulin resistance.
Also, I would suggest including some references in these sentences.
3- In line 321: "Additionally, we excluded some studies involving diet, as higher metabolite intake may artificially increase BCAA levels in metabolomics(Wang et al. 2011)". I believe that should be mentioned in the inclusion and exclusion criteria.

Experimental design

no comment.

Validity of the findings

no comment.

---

## Round 0.2 · Minor Revisions

Dear Authors, we are almost there. Please check and revise point by point the comments kindly provided by the reviewer.

Reviewer 1 ·

Basic reporting

The revised manuscript titled “association of circulating branched-chain amino acids with risk of pre-diabetes: a systematic review and meta-analysis” was a meta-analysis study to elaborate if there has a difference in serum/plasma BCAA(Val,Leu,Ile) levels between healthy people and preDM. In this study, the authors conducted a though comprehensive literature search of 9 databases and finally enclosed 16 studies. After information extraction and synthesis, they found that both plasma and serum BCAAs in preDM were higher than those in healthy people. The authors also carried out sensitivity analysis and publication bias analysis. The study proved that preDM is associated with increased BCAA, which may imply potential research directions of DM, and has practical significance. The conclusion obtained is reliable with proper data analyzing.

Several comments mentioned before have been revised properly.

Experimental design

no comment

Validity of the findings

no comment

Additional comments

Some minor issues still need to be mentioned:
1. Line 57 Is BCKDH an abbreviation? Could you write the full name?
2. Line 58 Is BCKAS an abbreviation? Could you write the full name?
3. Line56-61 Please indicate the references of these sentences?
4. I still suggest to use preDM and healthy, not patient and control. So can you make some modify of the topmost title of Fig2-5?Also Line229 “controls” better be “healthy”
5. Line 290 “The findings from our experiment are limited to…” Here “experiment” is inappropriate
6. Please give some explanation about ○4and ○5 in the last column of Table 1, as well as number (%) or medians (90% range) in the description.
7. Line 323-324 “suggesting that circulating BCAA could be used as a supplementary diagnostic tool for PreDM.” Could you explain it specifically? How?

---

## Round 0.3 · Minor Revisions

Dear Authors, Thank you for your revised manuscript. The first reviewer is satisfied with your responses. Congratulations on that progress.

However, we kindly ask that you address the remaining points raised by the second reviewer. Please respond to each of the following comments individually and thoroughly:

Reviewer 2 Comments:

Basic Reporting:
The manuscript is generally clear and well-written. The stated objectives have been met, and the conclusions are consistent with the findings. However, the following points require clarification:

1. Inclusion and Exclusion Criteria: You mention that patients with type 1 diabetes mellitus (DM) were excluded. Could you clarify whether the included studies also involved participants with type 2 DM, in addition to those with prediabetes (PreDM)? If so, was the comparison limited to PreDM versus controls? Please confirm whether type 2 DM patients were excluded from your analysis.

2. Clarification of BCAA Effects (Line 299): The sentence beginning with “BCAA is frequently reported to mediate anti-obesity, anti-diabetic, and anti-insulin resistance effects…” appears to suggest that BCAAs have beneficial effects. However, based on the context, it seems you intended to convey that BCAAs may have deleterious effects in these conditions. Please revise this sentence for clarity and accuracy. Additionally, we recommend including relevant references to support this statement.

3. Dietary Studies and BCAA Levels (Line 321): You mention that studies involving diet were excluded due to the potential for artificially elevated BCAA levels from increased metabolite intake (Wang et al., 2011). This rationale should also be explicitly stated in the inclusion and exclusion criteria section for transparency.

**Language Note:** When you prepare your next revision, please either (i) have a colleague who is proficient in English and familiar with the subject matter review your manuscript, or (ii) contact a professional editing service to review your manuscript. PeerJ can provide language editing services - you can contact us at [email protected] for pricing (be sure to provide your manuscript number and title). – PeerJ Staff

Reviewer 1 ·

Basic reporting

The revised manuscript was a meta-analysis study to determine if there is a difference in serum/plasma BCAA (Val Leu Ile) levels between healthy people and preDM. In this study, the authors conducted a thorough, comprehensive literature search of 9 databases and finally included 16 studies. After information extraction and synthesis, they found that both plasma and serum BCAAs in preDM were higher than those in healthy people. The authors also carried out a sensitivity analysis and a publication bias analysis. The study proved that preDM is associated with increased BCAA, which may imply potential research directions of DM, and has practical significance. The conclusion obtained is reliable with proper data analysis.

The comments mentioned before have all been revised or answered properly.

Experimental design

-

Validity of the findings

-

·

Basic reporting

I don´t think authors saw my suggestions, as there were no changes in the topics I mentioned, and there are answers for only 1 of the reviewers in the response letter. Please find the suggestions I have made so I can reanalyze the paper.

Experimental design

-

Validity of the findings

-

---

## Round 0.4 · accepted · Accept

Dear Authors:

Thank you for submitting the revised version of your manuscript. The reviewer acknowledges that the manuscript has properly addressed the previous criticism.
I am happy to recommend the acceptance of your work.

·

Basic reporting

With the modifications made, the paper is now fit for publication.

Experimental design

The research question is well defined and was answered accordingly.

Validity of the findings

Data is relevant, and the conclusions are well stated.